# Elevated $CO_2$ degassing rates prevented the return of Snowball Earth during the Phanerozoic

Benjamin J.W. Mills[1,2], Christopher R. Scotese[3], Nicholas G. Walding[2], Graham A. Shields [4] & Timothy M. Lenton[2]

The Cryogenian period (~720–635 Ma) is marked by extensive Snowball Earth glaciations. These have previously been linked to $CO_2$ draw-down, but the severe cold climates of the Cryogenian have never been replicated during the Phanerozoic despite similar, and sometimes more dramatic changes to carbon sinks. Here we quantify the total $CO_2$ input rate, both by measuring the global length of subduction zones in plate tectonic reconstructions, and by sea-level inversion. Our results indicate that degassing rates were anomalously low during the Late Neoproterozoic, roughly doubled by the Early Phanerozoic, and remained comparatively high until the Cenozoic. Our carbon cycle modelling identifies the Cryogenian as a unique period during which low surface temperature was more easily achieved, and shows that the shift towards greater $CO_2$ input rates after the Cryogenian helped prevent severe glaciation during the Phanerozoic. Such a shift appears essential for the development of complex animal life.

[1] School of Earth and Environment, University of Leeds, Leeds LS2 9JT, UK. [2] Earth System Science, College of Life and Environmental Sciences, University of Exeter, Exeter EX4 4QE, UK. [3] Department of Earth and Planetary Sciences, Northwestern University, Evanston, IL 60201, USA. [4] Department of Earth Sciences, University College London, Gower Street, London WC1E 6BT, UK. Correspondence and requests for materials should be addressed to B.J.W.M. (email: b.mills@leeds.ac.uk)

Acombination of physical, paleomagnetic and geochemical evidence points to at least two long-lived, low-latitude glacial events during the Cryogenian period of the late Neoproterozoic[1, 2]. Ediacaran strata also show evidence of glaciation, and icehouse climates are well known from the late Ordovician, Permo-Carboniferous and late Cenozoic (Fig. 1), but none of these later episodes show ice caps extending significantly further equatorward than the present day[3]. No Snowball Earth events have occurred during the last 600 million years. A low-latitude glaciation during this time would have had severe consequences for the emerging terrestrial biosphere[1]: multimillion-year periods of global ice cover and extreme cold temperatures, followed by a $CO_2$ super-greenhouse after glacial termination, provide kill mechanisms on a much greater scale than those identified[4] during Phanerozoic mass extinction events.

The onset of the deep glaciations in the Cryogenian is most often linked to climate cooling following the draw-down of atmospheric $CO_2$. This draw-down may have been triggered by the breakup of the supercontinent Rodinia (~750 Ma) and the emplacement of the Franklin large igneous province (~720 Ma), which would amplify the hydrological cycle and chemical weathering as well as delivering nutrients that fuel organic carbon burial[5–7]. Both silicate weathering (and the resulting burial of carbonates) and the burial of organic carbon are sinks in the long-term carbon cycle[8], and are expected to reduce atmospheric $CO_2$, but many $CO_2$ draw-down events have occurred at other times without causing extensive glaciation. For example, the Franklin large igneous province is estimated to have been around one quarter of the size of the equatorially positioned ~200 Ma Central Atlantic Magmatic Province[9], which was also emplaced during a time of supercontinent breakup (see Supplementary Fig. 1). Furthermore, the largest ever global enhancement of chemical weathering is thought to have occurred during the Carboniferous (~360–300 Ma) with the establishment of deeply rooted forest ecosystems[8, 10]. This ~fourfold enhancement of chemical weathering fluxes is much greater than the global enhancements associated with events in the Cryogenian[5, 11], and the ensuing late Palaeozoic icehouse was substantial (Fig. 1), but was not a Snowball Earth event.

In order to replicate extremely low Cryogenian temperatures, biogeochemical models have previously assumed a $CO_2$ degassing rate similar to the present day[5, 11], which when combined with the Neoproterozoic positioning of the continents, and the significantly lower solar flux, results in $CO_2$ less than six times preindustrial atmospheric level (PAL) and a global average surface temperature well below present day. This contrasts with long-term carbon cycle models for the early Paleozoic[10, 12], in which the degassing rate is inferred to be substantially higher

than present, and which predict high $CO_2$ concentrations (>10 PAL), supporting a global average surface temperature above 20 °C. The $CO_2$ degassing rate is not the only control on surface temperature in these models (for example, palaeogeography plays an important role by controlling the efficiency of continental weathering[13]) but it is clearly a key factor in determining planetary temperature.

Recent studies have provided some evidence for an increase in the tectonic $CO_2$ source between the Cryogenian and early Palaeozoic: high proportions of detrital zircons with ages close to the host rock depositional age are indicative of continental arc systems at convergent plate margins[14], and U-Pb zircon age probability distributions based on this metric predict a marked increase in continental arc activity between the Cryogenian and Cambrian periods, and corresponding increased rates of $CO_2$ degassing[15]. A recent compilation of zircon measurements over the Phanerozoic shows a continued link between arc volcanism and warm climates[16], as do reconstructions of the spatial extent of continental arcs from the exposure of granitoid plutons[17]. But direct quantification of total $CO_2$ input rates is not possible via these methods, limiting our ability to view the state of the whole Earth system, which critically also depends on the linearly increasing solar flux over geologic time, and the relative efficiency of continental weathering in a world without land plants.

One way to quantify the tectonic rate of $CO_2$ input is through comparison to the length of subduction zones. Assuming relatively small long-term (~100 Myr) changes in plate velocity during the last 750 Myr (e.g., a velocity of between around 4 and 6 cm/yr)[18–20], total subduction zone length has been used to quantify the global $CO_2$ input rate via both arc and ridge degassing, as it approximates the total crustal addition and destruction rates[21]. The utility of subduction zone length as a $CO_2$ input proxy has been demonstrated for the Mesozoic and Cenozoic, where the reduction in the length of continental arcs likely contributed to cooling[22]. Using total subduction zone length as a $CO_2$ input proxy in biogeochemical models for the Mesozoic has been shown to resolve a long-standing mismatch between modelled $CO_2$ concentration and that inferred from the long-term trend in proxy data[21].

The PALEOMAP Project has produced global plate tectonic reconstructions showing the past configuration of the continents and ocean basins[23], including the location of past plate boundaries[24]. The most recent maps[25] illustrate the global pattern of evolving plate boundaries for the past 750 million years (e.g. Fig. 2). These maps are produced by defining rules for plate motion that specify movement away from spreading centres and towards trenches. Plate boundaries are then added that describe the locations of active mid-ocean ridges, strike slip boundaries and subduction zones necessary to produce the observed plate motions. The missing ligations are inferred because subduction zones must form long, continuous plate boundaries, and mid-ocean ridges are inferred to have been located in oceans where the plates were moving apart. This approach involves significant uncertainty (see Supplementary Fig. 2), and the accuracy of these subduction zone maps depends on four independent lines of evidence: direct observation of volcanic and plutonic rocks, imaging of subducted slabs in the mantle for the Mesozoic and Cenozoic[21, 26], the timing of collisions and terrane accretion and analysis of relative plate motions and plate boundary reconstructions[27, 28].

In this paper, we attempt to quantify the rate of $CO_2$ input to the surface system from the Cryogenian to present day, by measuring the total global length of subduction zones in PALEOMAP plate tectonic reconstructions.

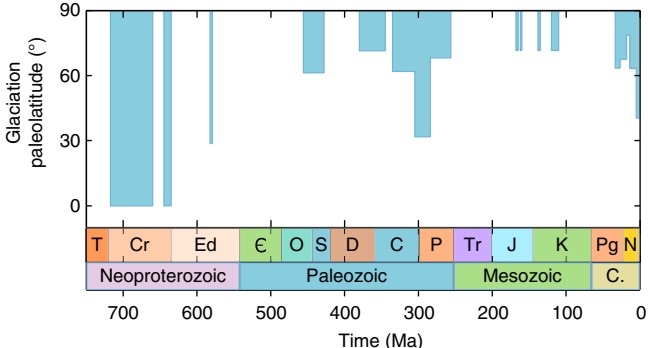

**Fig. 1** Paleolatitude of glaciations throughout the Neoproterozoic and Phanerozoic. Although there are regular icehouse periods[3], ice cover has not reached the tropics since the Cryogenian (720–635 Ma)

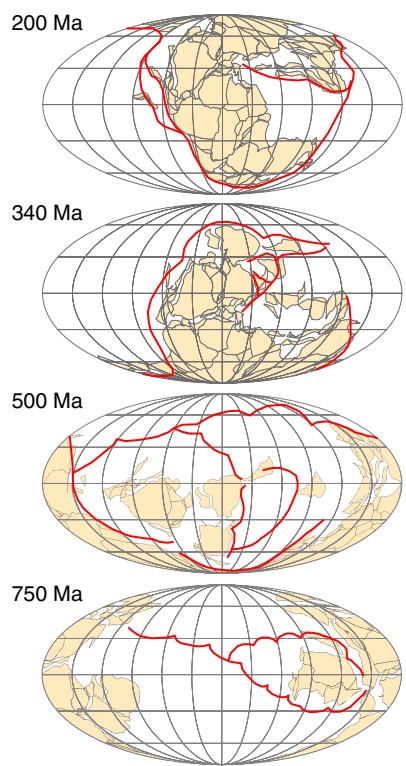

**Fig. 2** Past subduction zones shown in simplified PALEOMAP reconstructions. Subduction zones in red. See 'Methods' and Supplementary Fig. 3 for more information

## Results

**Subduction zone lengths.** Figure 3a shows the total global subduction zone length for 750 Ma to present, calculated from 33 PALEOMAP reconstructions (black line), with error estimates shown in grey. For completeness, we also calculate the total length of subduction zones from Colorado Plateau Geosystems' Deep Time Maps[TM], which use a similar combination of techniques to the PALEOMAP project, although this implies similar uncertainties, e.g., due to lack of evidence for ancient intraoceanic subduction. These results are compared to the global subduction zone lengths of Van Der Meer et al.[21], which are derived for the Mesozoic and Cenozoic from mantle tomography, and Meredith et al.[18], which are from kinematic modelling. Images of individual plate tectonic reconstructions are included in Supplementary Figs. 3 and 4, see 'Methods' for further discussion and error analysis.

Our calculated total subduction zone lengths identify the late Neoproterozoic, late Palaeozoic and present day as times of minimal subduction zone length, with sustained higher lengths during the early Palaeozoic and Mesozoic. This pattern is similar to previously inferred variations in continental arc activity and lengths made via the abundance of young zircon grains[16] and the surface exposure of granitoids[17], respectively (Fig. 3b). Our results for total global lengths are not expected to agree directly with the calculated lengths or activities of continental arcs alone, but may be expected to show a similar trend, as both are driven to some degree by ridge generation rates (see Supplementary Fig. 5 for slab flux comparison).

The clearest available test of our metric for $CO_2$ input is comparison to reconstructions of global ridge generation. Sea-level inversion[29] has used to infer ancient ridge volumes and generation rates, and is used to drive the $CO_2$ degassing rates in the widely used biogeochemical models GEOCARB[30] and COPSE[31], but it has not been extended to paleo sea-level data

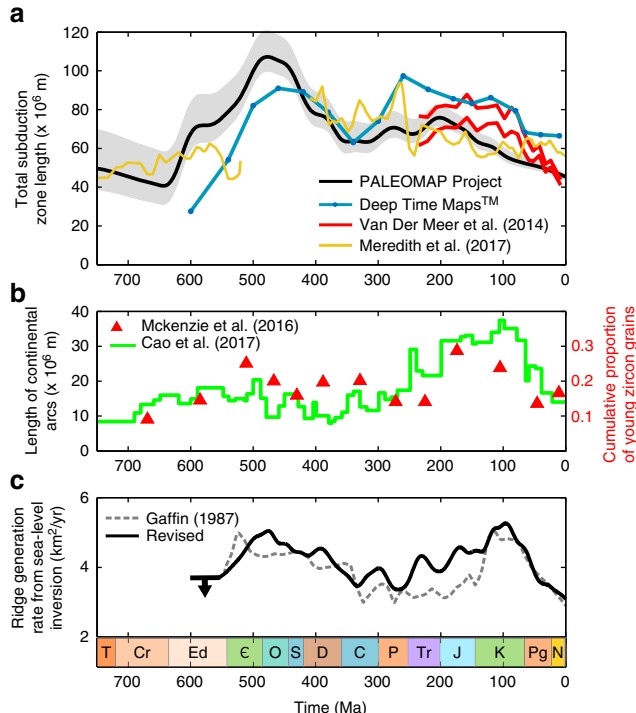

**Fig. 3** Subduction zone lengths. **a** Total global subduction zone length derived here from the plate tectonic reconstructions of the PALEOMAP project and from Deep Time Maps[TM], compared to lengths derived from seismic tomography (Van Der Meer et al.[21] shown in red) and from kinematic modelling[18]. Error estimations (grey) based on observations, direct plate imaging, timing of collisions and relative plate motions (see 'Methods' and Supplementary Fig. 2). **b** Global length of continental arcs[17] (left axis) and the cumulative proportion of young Zircon grains[16] (shown in red, right axis). **c** Ridge generation rate via revised sea-level inversion method[29], original predictions shown as grey dashed line, revised predictions (see text) shown as black line. See 'Methods' for full details

sets beyond the initial 1977 data of Vail et al.[32]. We construct here a simplified version of the sea-level inversion method (which ignores hypsometry) to estimate ridge generation rates from more recent paleo sea-level compilations[33, 34]. Our method closely approximates the results of the original derivation when applied to the original data (see Supplementary Figs. 6 and 7).

Ridge generation rates from the original and revised sea-level inversion are plotted in Fig. 3c. The method carries its own uncertainties[35], and cannot be extended directly into the late Neoproterozoic due to lack of data, but the Phanerozoic results agree well with the trends and magnitudes of variation shown in the plots of total subduction zone length (Fig. 3a), and are also similar to ridge generation patterns inferred from a strontium mixing model[36]. Sea level in the Ediacaran period was most likely lower than during the Cambrian[37], and adding this assumption to the model results in ridge generation rates that are also lower (black arrow in Fig. 3c). We conclude from these calculations that combined tectonic $CO_2$ input rates from arc and ridge degassing most likely followed a 'double-humped' curve over the Phanerozoic, with broad peaks during the early Palaeozoic and Mesozoic. Moreover, the correspondence between ridge generation rate and total subduction zone length implies that there were no long-term (>100 Myr) secular changes in plate velocity, and that therefore the record of total subduction zone lengths is a reasonable quantitative proxy for total $CO_2$ input.

Of course, uncertainties remain in both the total length of subduction zones, and in the link between this metric and the

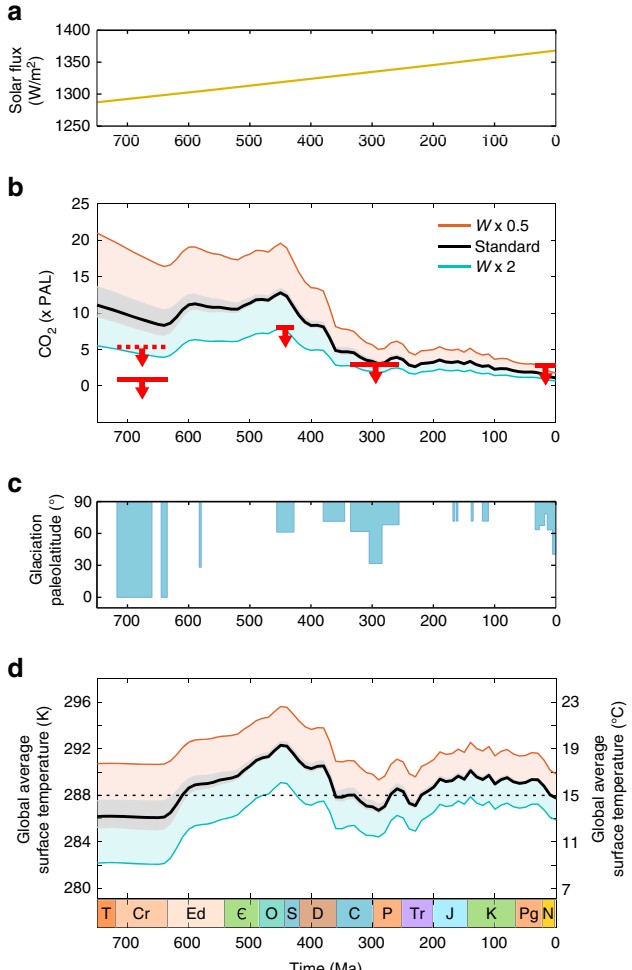

**Fig. 4** Model predictions. Model of Mills et al. (ref. [40]), with relative degassing rate set proportional to PALEOMAP subduction zone length. **a** Model solar flux. **b** Relative atmospheric $CO_2$ concentration shown against suggested thresholds for glaciation (red bars, see text). **c** Glaciation paleolatitude, as in Fig. 1. **d** Modelled global average surface temperature. For model runs, black lines show standard run, as in the COPSE model (see text), blue lines show twofold enhancement of terrestrial weathering ($W = 2$), representing a lower bound under potential glaciation trigger events. Orange lines show a twofold suppression of weathering rates ($W = 0.5$) for contrast

global $CO_2$ degassing rate. Kinematic plate models[27, 38, 39] represent the state of the art in reconstructing global subduction mechanics, and have recently been attempted for the Neoproterozoic[18]. The total length of subduction zones in these models[18, 39] is plotted against our derived lengths in Fig. 3a. Although this approach has not yet been linked between the Cambrian and Devonian periods, the available reconstructions show broad agreement on the length of subduction zones. Long-term (~100 Ma) average model plate speeds for the Neoproterozoic are around 4 cm/yr, reasonably similar to the present, but may have been higher in the Paleozoic[39], and models for the Mesozoic show shorter periods (~10–20 Myrs) of significantly higher speeds[38], and imply a more complex and varying degassing rate over these timescales, which may not correspond to large changes in subduction zone length. In addition, continental arc magmatism involves the heating of crustal carbonates, potentially increasing the $CO_2$ source above that occurring from subduction at oceanic arcs[16, 22].

Despite these uncertainties, all of the available data points to a significant increase in long-term $CO_2$ input rates between the Cryogenian and the Palaeozoic, and our measurements of subduction zone lengths provide a novel mechanism for quantifying this increase. We now explore the likely effects on atmospheric $CO_2$ concentration, global average surface temperature and the relative ease with which a glaciation can be initiated.

**Biogeochemical modelling.** Global average surface temperature depends on the incoming solar flux, planetary albedo and the radiative properties of the atmosphere (i.e., greenhouse gases). Here we use a recently published steady-state biogeochemical model[40] (a simplification of COPSE[31]) to assess the combined impact of our predicted changes to $CO_2$ outgassing, and the steadily increasing solar flux. Albedo in the model is calculated as a function of surface temperature, and the carbon cycle follows the GEOCARB models[10, 30] (see 'Methods' for more details). The $CO_2$ degassing rate in the model is scaled to the total subduction zone length following previous work[21]. Model output for relative atmospheric $CO_2$ concentration and global average surface temperature is shown in Fig. 4 alongside the severity of known glacial periods. Grey error bars on model outputs follow the errors estimated for the PALEOMAP subduction zone length input.

Model predictions shown in black demonstrate the model 'baseline conditions'—here the model includes only the most basic changes to weathering fluxes: an assumed increase in terrestrial weatherability coincident with the evolution of rooted plants[8], and a reduced erosion rate during the mid Phanerozoic[8], as in the original COPSE model[31]. The model does not include a number of potential changes to the silicate weathering $CO_2$ sink (e.g., due to LIP emplacement, the supercontinent cycle and palaeogeography, or uplift and exposure of different lithologies[5, 6, 41, 42]), which might be associated with individual icehouse events, but are uncertain, and often difficult to quantify at the global scale. The potential for such changes in $CO_2$ sinks to drive cooling is summarised by the blue model output, which plots model steady states under a continuous twofold enhancement of terrestrial weathering fluxes, comparable to the proposed proximate triggers for glaciations throughout the Neoproterozoic[5, 6] and Paleozoic[12]. The intention here is to show a reasonable lower bound for $CO_2$ and temperature in the model, which shows the potential to reach low temperature for shorter periods (<100 Myrs) comparable with the establishment of ice caps.

Model $CO_2$ predictions follow the shape of the subduction zone length curve, as this controls the rate of $CO_2$ input, but this is imprinted on a general long-term decline in $CO_2$ concentration due to the rising solar constant and establishment of rooted plants, which both act to enhance silicate weathering. Additional transient weathering enhancements (demonstrated by the $W = 2$ line) are capable of driving low global average temperatures compatible with low-latitude glaciation (e.g., ~10 °C, ref. [43]) during the Cryogenian, but the same enhancements have less effect during the Ediacaran and Palaeozoic. It is possible to drive surface temperatures close to, or lower than present day (i.e., compatible with the Hirnantian icehouse around 440 Ma), but not to the extent where ice-albedo runaway would lead to Snowball Earth conditions. By the time that the reconstructed $CO_2$ degassing rate begins to fall in the mid-Palaeozoic, the solar flux has increased sufficiently to result in baseline surface temperatures similar to the present day. The continual rise of the solar flux throughout the remainder of the Phanerozoic, coupled with higher-than-present rates of $CO_2$ degassing, ensures that background surface temperature is never low enough to allow for weathering enhancements to cause low-latitude

glaciations. See Supplementary Fig. 8 for model sensitivity to $CO_2$ degassing rates.

Although illustrative, global average surface temperature is not the sole parameter controlling ice sheet advance, which will operate differently depending on the position of the continents and dynamics of the ocean. We therefore also plot suggested $CO_2$ thresholds for individual glaciations[5, 44–47] against our modelled $CO_2$. In general, the lower bound for $CO_2$ concentration in the model (e.g., assuming some weathering enhancement, blue line) is below these thresholds. The $pCO_2$ and global average surface temperature required to initiate a Snowball Earth glaciation in the Neoproterozoic varies significantly between models[44] with more complex spatially resolved models requiring lower concentrations of $CO_2$. In our simple non-spatial model, achieving global average surface temperature below 10 °C is very difficult, due to powerful global negative feedback[31, 48]. However, it has already been shown that a Neoproterozoic Snowball Earth can be achieved in a spatial model under a sensible enhancement of weathering fluxes, providing that the $CO_2$ outgassing rate is similar to present[5], providing support for our hypothesis.

## Discussion

We have attempted to quantify tectonic $CO_2$ input rates over the past 750 million years by measuring subduction zone lengths and assuming a constant global average plate velocity. Accurately reconstructing plate tectonics in the Neoproterozoic is difficult, and hampered by a lack of intra-oceanic subduction data. In addition to this, changes in plate velocity, as well as any changes in the carbon content of the crust, and the potential for continental arcs to release additional crustal carbon, will also contribute to the global $CO_2$ degassing rate. Furthermore, it is uncertain how much carbonate has been subducted through time, how much slab carbonate undergoes decarbonation and how this amount is related to net $CO_2$ production[49]. In the modelling here, we have assumed that the rate of $CO_2$ release corresponds directly to the rate of material subduction.

It is likely that the carbon content of the crust has generally increased over time[50, 51], supporting our conclusions, but the composition of subducted material remains a key uncertainty. Variations in plate velocity may add further complexity to these scenarios, although the long-term (~100 Myr) rates suggested by thermal evolution modelling[19, 20] do not change dramatically over this timeframe. We have explored additional model uncertainty in the SI that shows the sensitivity to changes in degassing rate.

To summarise our results, $CO_2$ degassing rates appear to have increased twofold between the Cryogenian and early Palaeozoic, and remained reasonably high until the Cenozoic. When combined with the increasing solar flux, the Cryogenian emerges as a unique period of extremely low background temperature, in which the Earth could have easily been pushed into global glaciation under moderate enhancements of carbon sinks due to weathering and biological events[5–7]. Such enhancements would fail to produce comparable extreme low temperatures during the early Phanerozoic, potentially explaining why Palaeozoic weathering events, whether attributed to tectonics[52] or the biosphere[10, 12] did not lead to a return to Snowball Earth—a condition that would have been catastrophic for the continued development of complex life. While there is strong evidence for weathering and carbon draw-down events preceding many glacial periods, it is clear that variable rates of $CO_2$ degassing play an important role. Continued effort is required to accurately reconstruct both ancient plate tectonics and enhancements to carbon burial fluxes.

## Methods

**PALEOMAP reconstructions.** For the past 40 years, the PALEOMAP Project has produced global plate tectonic reconstructions showing the past configuration of the continents and ocean basins[53–55]. Some of these maps include the location of past plate boundaries[24, 56–58]. The most recent set of maps[25] is the first publication to illustrate the global pattern of evolving plate boundaries for the past 750 million years.

The accuracy of these subduction zone maps depends on four independent lines of evidence (Supplementary Fig. 2). The first is the direct observation of volcanic and plutonic rocks produced by subduction activity and sediments that formed in accretionary prisms. The amount of direct geological evidence is excellent for the Cenozoic and Cretaceous, but decreases linearly back into the Palaeozoic and Precambrian. At 300 Ma, only 50% of the volcanic, plutonic and accretionary evidence is still preserved. At 750 Ma, less than 15% of the geological evidence of past subduction is still preserved.

The second line of evidence is the direct imaging of subducted slabs through mantle tomography. Seismic waves travel faster through the colder, denser subducted slabs and hence, the slabs appear as velocity anomalies. Because all subducted slabs are eventually assimilated back into the mantle as they sink into ever warmer regions, there is a maximum age that can be imaged. The maximum age of imaged slab graveyards is estimated to be 130–300 Ma[26, 59].

The timing of continental collisions and terrane accretion is the third line of evidence that can be used to map ancient subduction zones. This follows from the simple logic that the 'sutures' that mark the collision zones between continents were once ancient subduction zones. The polarity of subduction can be determined by vergence of thrust belts and the presence of ophiolitic remnants of subducted ocean floor. The formation of the supercontinents: Pannotia (~600 Ma), Pangea (~300 Ma) and Eurasia (~50 Ma), increase the accuracy with which we can map the antecedent ancient subduction zones.

When all other lines of evidence are unavailable, it is still possible to predict the location of ancient subduction zones by carefully assessing relative plate motions. Simply put, if two plates are moving towards each other then there must be an intervening subduction zone. Conversely, if two plates are moving apart, then there must be a mid-ocean ridge between them. By carefully analysing the plate motions through time, it is possible to assemble a parsimonious model of evolving subduction zones and spreading ridges that explains these plate motions[60]. For examples of continuously evolving plate boundary reconstructions, see refs. [27, 28].

These models of evolving plate boundaries help to map ancient subduction zones for the times when more direct geological and geophysical evidence of subduction is lacking. High-quality plate tectonic models can be made for times since the early Cretaceous (130 Ma) due to the availability of preserved ocean floor, hot spot tracks and abundant, high-quality paleomagnetic data[61]. The quality of paleomagnetic data prior to 300 Ma, decreases drastically, reducing the quality of the plate models ('world uncertainty'). Prior to the base of the Cambrian, the lack of fossils makes accurate time correlation very difficult, hence Precambrian plate models are somewhat speculative.

The location of subduction zones can be mapped with high degree of confidence back to the late Cretaceous, ~70 Ma. Between 70 and 130 Ma, there is still a sufficient amount of information to provide a good level of confidence. From 130 to 400 Ma, reasonable estimates of the location of subduction zones can be made because this was the time of the formation of the supercontinent, Pangea. Similarly, the location of subduction zones associated with the formation of the late Precambrian supercontinent, Pannotia, formed ~600 Ma, can be identified with some accuracy. Informed, but necessarily speculative maps can be made showing the location of ancient subduction zones after the breakup of Rodinia (750–650 Ma[24]) and for the time interval between 550 and 400 Ma. Although intra-oceanic subduction may be underestimated due to the lack of preserved record, such as in the Mesozoic Tethys and Panthalassa Oceans.

**Sea-level inversion.** The below method is a simplified version of the computations made by Gaffin[29] to determine ridge production rates from sea-level changes. It produces almost identical results when applied to the sea-level data of Vail et al.[32], which was used in Gaffin's study. We apply the method to a recent sea-level data set[33, 34]. The results are shown in Supplementary Fig. 6.

We assume that ridge volume (RV) can be calculated from total volume and water volume using the following expression (see Eq. 20 in ref. [29]):

$$\text{RV} = A_{\text{ocean}}(d_{\text{present}} + \Delta\text{d}) - V_{\text{water}} \qquad (1)$$

It is then assumed that changes in ridge volume correspond to production and destruction:

$$\frac{\text{d(RV)}}{\text{d}t} = R_{\text{prod}} - R_{\text{dest}} \qquad (2)$$

Where the destruction rate is assumed to increase linearly with total ridge volume:

$$R_{\text{dest}} = k_{\text{dest}}\frac{\text{RV}}{\text{RV}_0} \qquad (3)$$

Where $k_{\text{dest}}$ is the present day rate, calculated to recover the present day rate of production. We then solve for $R_{\text{prod}}$ using numerical differentiation to obtain $\frac{\text{d(RV)}}{\text{d}t}$

Parameters used follow Gaffin[29]:

$A_{ocean} = 360 \times 10^6$ km, $d_{present} = 5.4$ km, $V_{water} = 1.75 \times 10^9$ km$^3$, and we take $k_{dest} = 3.5$ km$^2$/yr.

We ignore many of the nuances of Gaffin's work here, such as isostasy and hypsometry, to simplify the mathematics. But the predictions are very similar, owing in part to the close correspondence between the predicted ridge production rate and the sea-level curve input. Also, it should be noted that while ridge volume contributes greatly to sea-level changes, there are other significant contributions[35] (e.g., climate).

Quantitative eustatic sea-level estimates are unavailable for the Neoproterozoic. However, sea level was most likely lower during the Ediacaran than during the Early Cambrian[37]. Supplementary Fig. 7 shows the inversion model run with a fixed lower Cambrian sea level for the period 600–550 Ma, representing the likely maximum.

**Biogeochemical modelling method.** The model used in this work is described in detail in ref. [40], and represents the global long-term carbon cycle, calculating atmospheric $CO_2$ concentration and temperature at steady state for variations in the volcanic $CO_2$ source, solar flux and potential enhancements to terrestrial silicate weathering. The model is a simplified version of the COPSE model[31], which itself is similar to the GEOCARB models[10].

The only model alteration is the revision of the seafloor weathering flux (low-temperature basalt alteration) to include direct temperature dependence, instead of a dependence on atmospheric $CO_2$ concentration, which is more realistic. The revised flux assumes a direct relationship between surface temperature change and seafloor temperature:

$$sfw = k_{sfw} \cdot D \cdot e^{0.061(T-T_0)} \qquad (4)$$

Here $k_{sfw}$ is the present day rate, set as $1.75 \times 10^{12}$ mol/yr, $D$ is the relative degassing rate, $T$ is global average surface temperature and $T_0$ is present day average surface temperature. This alteration makes very little difference to the results shown in this paper, but is applied for completeness and constancy with other work using versions of this model[62]. In this paper, we vary the model degassing flux, assumed to be proportional to the total subduction zone length, as explored by Van Der Meer et al.[21]. Model equations are shown in Supplementary Note 1.

During the Phanerozoic, the model includes 'forcings' that affect the rate of terrestrial weathering. The dominant of these is a fourfold enhancement of silicate and carbonate weathering coincident with the evolution of rooted plants around 400 Ma. The model also includes a varying rate of erosion with a minimum in the early Mesozoic. Additional forcings have been added in the GEOCARB models, and in some extensions of COPSE[63], these include changing lithology (including volcanic weathering), an approximation of the changes in runoff due to changing palaeogeography, and changes to climate sensitivity. We do not include these extensions here and use the baseline COPSE model forcings of uplift (U) and biological weathering (W) for the Phanerozoic. We do not assume any forcings for the Neoproterozoic, and for the whole model timeframe, we use the degassing curve derived in this work instead of that used in COPSE.

**Data availability.** A data table for the subduction zone length curve derived here, as well as model code and outputs are available from B.J.W.M. (b.mills@leeds.ac.uk).

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

## Acknowledgements

B.J.W.M. is funded by a University of Leeds Academic Fellowship. B.J.W.M. and T.M.L. were also supported by the Leverhulme Trust (RPG-2013-106), G.A.S. and T.M.L. were also supported by NERC (NE/P013651/1) and (NE/P013643/1). We thank R. Blakey for information on the construction of Deep Time Maps.

## Author contributions

B.J.W.M. designed the study and computed sea-level inversions, N.G.W. and B.J.W.M. computed subduction zone lengths with help and additional analysis from C.R.S. B.J.W.M., T.M.L. and G.A.S. analysed the results and model output. B.J.W.M. and C.R.S. wrote the manuscript with input from all authors.

## Additional information

**Competing interests:** The authors declare no competing financial interests.

7