## [Peer Review File · Nature Communications]

Reviewers' comments:

Reviewer #1 (Remarks to the Author):

This is an important paper. Authors propose that the lengths of subduction zones have increased in the Phanerozoic, leading to enhanced CO₂ production. They suggest that enhanced CO₂ production during the Phanerozoic may have kept the Earth out of Snowball events and that low levels of CO₂ production in the neoproterozoic set Earth's baseline climate to be sensitive to snowball events.

This paper resolves the fundamental problem the entire community has been having regarding Snowball Earth events and more. Standard view of Snowball events is that the efficiency of CO₂ drawdown, either through weathering or organic C deposition, was higher during that time, because of the generation of a large igneous province. However, LIPS have been around at other times of Earth's history, including the Phanerozoic, but no global glaciation - snowball - occurred.

In summary, they argue that the global long term CO₂ input baseline has increased with time, through an increase in subduction zone length. I think this is an important finding and worthy of publication in Nature Communications.

I think most of their analyses are robust, though I have some criticisms on small components of their paper, which in theory should be straightforward to address.

My main comments are as follows.

1) It would be ideal if the authors explained in more detail how they actually calculate subduction zone length from continent reconstructions back to the Paleozoic. What are the uncertainties? I also note that in the SI, the authors suggest they are using seismic "anisotropy". I assume this is a mistake and that they meant seismic tomography.

2) The authors use Vail/Haq sea level curves to estimate mid-ocean ridge spreading rates, under the assumption that enhanced spreading gives rise to higher sea level due to buoyancy effects of the lithosphere. I don't think this analysis is essential to their argument. It might be better to drop this analysis. For one, the Vail/Haq sea level curve is based on relative "sea level" and the curves come from sequence boundaries. But many things control sequence boundaries - sediment supply, accommodation space, glacial eustasy, mor spreading, etc. Dynamic subsidence also plays a key role at certain times. So to equate the sea level curves to mor spreading is an oversimplification, and probably will turn out to be wrong in the future.

3) The authors discuss near the end the possibility of the crustal carbon reservoir increasing through time. It may be worth citing or examining the papers by Lee et al. Nature Geoscience 2016 on the rise of atmospheric oxygen. In that paper, they show explicitly how the crustal carbon reservoir increases with time, which in turn leads to a steady increase in the volcanic/metamorphic production of CO₂ (and ultimately O₂ production). They also suggest that greenhouse conditions may be more prevalent in the Phanerozoic.

4) In modeling pCO₂, the authors might also want to take a look at Caves et al. in EPSL 2016. This paper discusses different sensitivities of weathering and the extent to which pCO₂ can be "driven" by variations in the efficiency of the sink. In a contrasting paper, Lee et al. in EPSL Frontiers 2015 show how arc volcanism flare-ups predict greenhouse conditions, followed by icehouse (not snowball) conditions, a feature characteristic of the Phanerozoic.

5) Fundamentally, the authors are saying that the C cycle is driven by the long-term inputs with weathering as a modulating feedback. I could not agree more. However, it may be worth highlighting other papers to this effect. For example, Caldeira and Berner in Geology a number of

years back addressed this issue.

6) Biogeochemical modeling. Since there is a SI, please provide more details.

Reviewer #2 (Remarks to the Author):

In their contribution, the authors suggest that very low Neoproterozoic CO₂ outgassing rates (as constrained by the global length of subduction zone in plate tectonics reconstructions) may constitute a previously unrecognized mechanism to explain the severity of the Neoproterozoic glaciations. The paper is very well written and the point the authors want to reach is particularly straightforward. While for years, a part of the community has been focused on better constraining the CO₂ sinks for the long-term carbon cycle, these last years have seen more and more papers putting forward the forgotten role played by the Solid Earth outgassing (van der Meer et al., 2014, PNAS, McKenzie et al., 2016, Science). This paper follows the same trend and resembles to a synthesis paper of the current state of our knowledge of the Earth degassing rate.

The main drawback of this approach remains the way we need to go from geometric calculations (i.e. subduction, ridge lengths, continental arc surfaces) to an estimate of the CO₂ outgassing. It is generally assumed that the degassing rate is proportional to the subduction length. As recognized by the authors, the nature of the sediments that were subducted may influence the rate of CO₂ outgassed by units of length. This may modulate the relationship between the subduction length and the degassing rate. Is this factor of secondary or primary importance when calculating atmospheric CO₂ levels remains an unsolved question.

Another annoying point is the way the authors are putting in opposition the effect of the continental drift on the CO₂ sink through continental weathering modulation and the effect of the CO₂ outgassing rate. It is written in some places that the continental break-up is a short term forcing – this is clearly not true. The time scale on which continental drift occurs is on the same order than the one on which Earth degassing rate is modulated, as they are both linked. It is the game when publishing in Nature journals to make our message clear and unique but still, it is desirable, when possible, not to oversell our results.

Here are some sentences/paragraph that should be modified to keep the paper with a more balanced view of processes at stake:

l. 51-52, ... However, supercontinent break-up is not unique to the Neoproterozoic,...

I don't like this kind sentence. Why ? Because, yes, other supercontinent break-up has occurred over the Phanerozoic but none of those younger supercontinent break-up has occurred across a low latitude ring, as it was the case for the Neoproterozoic. And this is that particular latitudinal position of the Rodinia supercontinent that explains why the break-up has resulted in very cold climatic conditions. Godderis et al. (2014, ESR) have quantified the effect of the younger continental break-up.

l.61-70, the authors start with a false statement. Studies are not assuming low background CO₂ concentrations for the Neoproterozoic. This low CO₂ concentration has been found using a complex climate-carbon model showing that the general cool climate of the Neoproterozoic is the result of the low latitude position of the continents, optimizing CO₂ consumption through silicate

weathering. There is no contrast with the Paleozoic where carbon cycle models found CO₂ concentrations that are more than double. Once again, this is the result of the continental configuration where most continents were localized in the middle and the high latitudes of the South Hemisphere. Indeed, such a position has been shown to generate larger CO₂ levels, reaching more than 20 PAL (Nardin et al., 2011, GSAB, Godderis et al., 2014, Earth Science Reviews)

then the authors write : If both of these model scenarios are correct, then some critical difference must have existed between the Cryogenian and the following periods of Earth history, which permitted very low background CO₂ concentrations in the former.

Yes, one fundamental difference is the position of the continents at this time (low latitudes) that has been unique in the last 800 Myrs. I don't want to reject the paper on this ground. But still, I think that the authors should be more balanced. It appears that it is the convergence of two processes that may have explained low latitude glaciations: both large continental weathering and very low Earth degassing. It is not black or white !

l. 165-169... but do not include any potential shorter-term enhancements to the silicate weathering CO₂ sink (e.g. due to LIP emplacement or supercontinent breakup) which might be associated with individual icehouse events.

The break-up of Pangea starts at 220-200 Ma and reaches its paroxysm at the end of the Cretaceous (65 Ma). I don't see any reasons to call this short-term event.

Figure 4B-C and l. 170-183, the approach taken here is really simple. They increase by a factor of two the CO₂ sink to mimic ? LIP emplacement or supercontinent break-up ? for a very long time period from 750 to 400 Ma. It makes no sense. For a supercontinent break-up, processes occur over several Myrs but are neither instantaneous nor linear. They can be punctuated by non-linearity (see Godderis et al., 2008, EPSL, Kent and Muttoni, Clim Past). The authors then use the mean global temperature to infer a potential for glaciation or not of their simulation. Here again, I am not really convinced. The Antarctica glaciation occurs while the atmospheric CO₂ level was around 800 ppm (Gasson et al., Clim Past) and the mean global temperature well above the pre-industrial value. Northern hemispheric glaciation occurs below a CO₂ threshold of 280 ppm with global temperature reaching 14 °C. If the authors want to use their global temperature, they should, at least, use the studies published on the CO₂ thresholds for each glacial event (several exist for the Neoproterozoic, Pollard, Donnadieu etc... for the Ordovician, you can use studies by Pohl et al. and for the LPIA, studies by the group of Chris Poulsen) and find the corresponding mean global temperature.

A last point, why did you plot CO₂ and mean T for the time period 750 - 400 Ma while your reconstruction has been done for the period 750 - 0 Ma ? DO I have missed something or please state the reasons behind this particular choice.

Please see attached file for more comments

Reviewer #3 (Remarks to the Author):

The authors aim to explain why there has not been a return to a Cryogenian Snowball Earth climate during the Phanerozoic. This is argued done two-ways, via global length of subduction zones from a plate tectonic reconstruction and via sea-level inversion. As such the application of these methods this far back in time is novel and the paper potentially provides a key insight in paleo-climate of the Neoproterozoic-Phanerozoic boundary which is original and will influence the thinking in the field. Neither method is perfect and are hindered by a lack of intra-oceanic subduction data and respectively, due the fact that paleo-sea-level is influenced by more than

factors just ocean spreading rates. More discussion on these limitations should be included. The paper would also benefit from the incorporation of more recent sea-level curves, and comparisons to plate tectonic activity from other plate tectonic models that also go back to this far back in time. Figure 4 should include a view on the entire Phanerozoic, providing evidence that the methodology works at times where more data is available.

Reviewer response for “Elevated CO₂ degassing rates prevented the return of Snowball Earth during the Phanerozoic”.

B. J. W. Mills et al.

We thank the reviewers for taking the time to give constructive comments on our manuscript, which have led to improvement of the work.

To summarise the alterations: figure 3 is updated to compare our subduction zone lengths to a recent publication which uses a different kind of model; figure 4 is updated to show the whole Phanerozoic and to include CO₂ thresholds for glaciation and the solar flux; the sea level curve has been updated, and the model equations are now shown in the SI. We have also modified several sections of the text and the citations to improve balance.

Due to these changes, and in light of the positive initial responses of all referees, we now believe that our paper is suitable for publication in Nature Communications.

A complete response to reviewers follows. Our responses are in blue text, as is any altered text in the accompanying revised manuscript.

Reviewer #1 (Remarks to the Author):

This is an important paper. Authors propose that the lengths of subduction zones have increased in the Phanerozoic, leading to enhanced CO₂ production. They suggest that enhanced CO₂ production during the Phanerozoic may have kept the Earth out of Snowball events and that low levels of CO₂ production in the neoproterozoic set Earth's baseline climate to be sensitive to snowball events.

This paper resolves the fundamental problem the entire community has been having regarding Snowball Earth events and more. Standard view of Snowball events is that the efficiency of CO₂ drawdown, either through weathering or organic C deposition, was higher during that time, because of the generation of a large igneous province. However, LIPS have been around at other times of Earth's history, including the Phanerozoic, but no global glaciation - snowball - occurred.

In summary, they argue that the global long term CO₂ input baseline has increased with time, through an increase in subduction zone length. I think this is an important finding and worthy of publication in Nature Communications.

I think most of their analyses are robust, though I have some criticisms on small components of their paper, which in theory should be straightforward to address.

My main comments are as follows.

1) It would be ideal if the authors explained in more detail how they actually calculate

subduction zone length from continent reconstructions back to the Paleozoic. What are the uncertainties? I also note that in the SI, the authors suggest they are using seismic "anisotropy". I assume this is a mistake and that they meant seismic tomography.

We have added a summarising paragraph to the main text on determining subduction zone lengths in PALEOMAP plate tectonic reconstructions, and the uncertainty involved. We hope this will make the process clearer to readers. We have tried to include as much as possible in the SI, but it gets to the point where it is better to consult the cited papers of Scotese et al. and the PALEOMAP webpage. We have also changed 'seismic anisotropy' to 'mantle tomography' as suggested: we agree this is a more appropriate term.

[ms line 96]

[SI line 26]

2) The authors use Vail/Haq sea level curves to estimate mid-ocean ridge spreading rates, under the assumption that enhanced spreading gives rise to higher sea level due to buoyancy effects of the lithosphere. I don't think this analysis is essential to their argument. It might be better to drop this analysis. For one, the Vail/Haq sea level curve is based on relative "sea level" and the curves come from sequence boundaries. But many things control sequence boundaries - sediment supply, accommodation space, glacial eustasy, mor spreading, etc. Dynamic subsidence also plays a key role at certain times. So to equate the sea level curves to mor spreading is an oversimplification, and probably will turn out to be wrong in the future.

We accept that the sea level method is an oversimplification, and we have now noted this more clearly in the text and added a citation to this effect. But we think that we still need to show how our results compare to this metric, especially as it is used in current modelling studies (e.g. Royer et al. 2014 Am. J. Sci.).

[ms line 136]

3)The authors discuss near the end the possibility of the crustal carbon reservoir increasing through time. It may be worth citing or examining the papers by Lee et al. Nature Geoscience 2016 on the rise of atmospheric oxygen. In that paper, they show explicitly how the crustal carbon reservoir increases with time, which in turn leads to a steady increase in the volcanic/metamorphic production of CO₂ (and ultimately O₂ production). They also suggest that greenhouse conditions may be more prevalent in the Phanerozoic.

We have now cited this paper

[ms line 224]

4) In modeling pCO₂, the authors might also want to take a look at Caves et al. in

EPSL 2016. This paper discusses different sensitivities of weathering and the extent to which pCO₂ can be "driven" by variations in the efficiency of the sink. In a contrasting paper, Lee et al. in EPSL Frontiers 2015 show how arc volcanism flare-ups predict greenhouse conditions, followed by icehouse (not snowball) conditions, a feature characteristic of the Phanerozoic.

We have cited these papers when referring to weathering CO₂ sinks. The uncertainty range for CO₂ and T we plot is aimed at broadly representing these ideas and other processes. We hope to explore the ideas in more detail in our future work.

[ms line 183]

5) Fundamentally, the authors are saying that the C cycle is driven by the long-term inputs with weathering as a modulating feedback. I could not agree more. However, it may be worth highlighting other papers to this effect. For example, Caldeira and Berner in Geology a number of years back addressed this issue.

We have cited this paper

[ms line 212]

6) Biogeochemical modeling. Since there is a SI, please provide more details.

We have added the full model equations to the SI. As stated, the model is unaltered from a previous publication (Mills et al. 2014, PNAS) aside from a minor update added in subsequent work, and the addition of the new degassing curve we wish to test.

[SI line 202]

Reviewer #2 (Remarks to the Author):

In their contribution, the authors suggest that very low Neoproterozoic CO₂ outgassing rates (as constrained by the global length of subduction zone in plate tectonics reconstructions) may constitute a previously unrecognized mechanism to explain the severity of the Neoproterozoic glaciations. The paper is very well written and the point the authors want to reach is particularly straightforward. While for years, a part of the community has been focused on better constraining the CO₂ sinks for the long-term carbon cycle, these last years have seen more and more papers putting forward the forgotten role played by the Solid Earth outgassing (van der Meer et al., 2014, PNAS, McKenzie et al., 2016, Science). This paper follows the same trend and resembles to a synthesis paper of the current state of our knowledge of the Earth degassing rate.

The main drawback of this approach remains the way we need to go from geometric

calculations (i.e. subduction, ridge lengths, continental arc surfaces) to an estimate of the CO₂ outgassing. It is generally assumed that the degassing rate is proportional to the subduction length. As recognized by the authors, the nature of the sediments that were subducted may influence the rate of CO₂ outgassed by units of length. This may modulate the relationship between the subduction length and the degassing rate. Is this factor of secondary or primary importance when calculating atmospheric CO₂ levels remains an unsolved question.

We have modified the discussion text to highlight this point.

[ms line 225]

Another annoying point is the way the authors are putting in opposition the effect of the continental drift on the CO₂ sink through continental weathering modulation and the effect of the CO₂ outgassing rate. It is written in some places that the continental break-up is a short term forcing – this is clearly not true. The time scale on which continental drift occurs is on the same order than the one on which Earth degassing rate is modulated, as they are both linked. It is the game when publishing in Nature journals to make our message clear and unique but still, it is desirable, when possible, not to oversell our results.

We have improved balance throughout and removed our use of ‘shorter term forcing’ when considering continental breakup, whilst trying to keep the message clear.

Here are some sentences/paragraph that should be modified to keep the paper with a more balanced view of processes at stake:

l. 51-52, ... However, supercontinent break-up is not unique to the Neoproterozoic,...

I don't like this kind sentence. Why ? Because, yes, other supercontinent break-up has occurred over the Phanerozoic but none of those younger supercontinent break-up has occurred across a low latitude ring, as it was the case for the Neoproterozoic. And this is that particular latitudinal position of the Rodinia supercontinent that explains why the break-up has resulted in very cold climatic conditions. Godderis et al. (2014, ESR) have quantified the effect of the younger continental break-up.

We agree with this point and have removed the text directly comparing the supercontinent breakups.

[ms line 47]

l.61-70, the authors start with a false statement. Studies are not assuming low

background CO₂ concentrations for the Neoproterozoic. This low CO₂ concentration has been found using a complex climate-carbon model showing that the general cool climate of the Neoproterozoic is the result of the low latitude position of the continents, optimizing CO₂ consumption through silicate weathering.

There is no contrast with the Paleozoic where carbon cycle models found CO₂ concentrations that are more than double. Once again, this is the result of the continental configuration where most continents were localized in the middle and the high latitudes of the South Hemisphere. Indeed, such a position has been shown to generate larger CO₂ levels, reaching more than 20 PAL (Nardin et al., 2011, GSAB, Godderis et al., 2014, Earth Science Reviews)

Our paragraph was poorly written and has been amended following this comment. What we wanted to say is that the ‘Snowball’ models (e.g. Donnadieu et al. 2004, Mills et al. 2011) assumed a present day CO₂ degassing rate for the Neoproterozoic, whilst many Paleozoic models used a significantly above-present degassing rate. We have amended the text in line with the above.

[ms line 59]

then the authors write : If both of these model scenarios are correct, then some critical difference must have existed between the Cryogenian and the following periods of Earth history, which permitted very low background CO₂ concentrations in the former.

Yes, one fundamental difference is the position of the continents at this time (low latitudes) that has been unique in the last 800 Myrs. I don't want to reject the paper on this ground. But still, I think that the authors should be more balanced. It appears that it is the convergence of two processes that may have explained low latitude glaciations: both large continental weathering and very low Earth degassing. It is not black or white !

We are completely in agreement with the above sentences. In our desire to draw attention to the contribution of degassing we have somewhat neglected the effects of continental position. We have now added citation to Godderis et al. (2014) as a good general reference, and modified text to generally improve balance throughout.

[ms line 65]

l. 165-169... but do not include any potential shorter-term enhancements to the silicate weathering CO₂ sink (e.g. due to LIP emplacement or supercontinent breakup) which might be associated with individual icehouse events.

The break-up of Pangea starts at 220-200 Ma and reaches its paroxysm at the end of the Cretaceous (65 Ma). I don't see any reasons to call this short-term event.

Our use of 'shorter term' was unfortunate. It was intended to distinguish between a transient (although long) event like continental breakup, and a permanent system change such as the evolution of rooted plants (enhanced weathering for the whole period from ~400 Ma - 0 Ma). We have now removed this text.

Figure 4B-C and I. 170-183, the approach taken here is really simple. They increase by a factor of two the CO2 sink to mimic ? LIP emplacement or supercontinent break-up ? for a very long time period from 750 to 400 Ma. It makes no sense. For a supercontinent break-up, processes occur over several Myrs but are neither instantaneous nor linear. They can be punctuated by non-linearity (see Godderis et al., 2008, EPSL, Kent and Muttoni, Clim Past).

We have revised the modelling plots and their description in response to this comment and others. The 'x2 weathering' line shows a lower bound of possible solutions if we enhance weathering, which is intended to show the possibility to cause low temperature over any desired timeframe by invoking either changes to paleogeography, LIP emplacement, or any of the great number of proposed 'glacial trigger' events in the literature.

The idea is that the reader can see what the same degree of perturbation might do at different points in Earth history. It is not intended that the reader follow the 'x2 weathering' line as a prediction of climate by itself - it is a lower bound. We have changed both the plot and text to make this more clear in the revision, and we thank the reviewer for encouraging us to do this.

[me line 178]

The authors then use the mean global temperature to infer a potential for glaciation or not of their simulation. Here again, I am not really convinced. The Antarctica glaciation occurs while the atmospheric CO2 level was around 800 ppm (Gasson et al., Clim Past) and the mean global temperature well above the pre-industrial value. Northern hemispheric glaciation occurs below a CO2 threshold of 280 ppm with global temperature reaching 14 °C. If the authors want to use their global temperature, they should, at least, use the studies published on the CO2 thresholds for each glacial event (several exist for the Neoproterozoic, Pollard, Donnadieu etc... for the Ordovician, you can use studies by Pohl et al. and for the LPIA, studies by the group of Chris Poulsen) and find the corresponding mean global temperature.

This is a good suggestion and we have extended the modelling results to show the whole Phanerozoic, and compared to published CO2 thresholds. As our model predicts CO2 we have shown the comparison directly instead of converting to temperature, as we believe this is clearer than trying to convert for each study/model. The comparison is reasonably good and the model 'lower bound'

passes all of the thresholds – with one exception for the Neoproterozoic snowball Earths: the range of CO₂ concentrations required in models for a Neoproterozoic ‘snowball’ glaciation depends greatly on the model complexity, and can be anywhere from 250-3000 ppm (Pierrehumbert et al. 2011). It seems that the lower of these estimates (from the more complex models) is more realistic, but our model is very simple, and designed for simulating climate similar to the present day, thus it cannot easily reach very low surface temperatures or very low pCO₂ in the Neoproterozoic because of the global negative feedbacks present.

However, the model of Donnadieu et al. (2004) shows that a snowball Earth can be triggered in a complex spatial model under a degassing rate similar to present – which is the rate predicted in our study for the Cryogenian. Therefore, even though our simple model cannot recreate the situation, it seems that our proposed degassing rate is sufficiently low to allow for a snowball glaciation, and therefore our conclusions are reasonable.

The text has been altered to include the above discussion.

[ms line 205]

A last point, why did you plot CO₂ and mean T for the time period 750 - 400 Ma while your reconstruction has been done for the period 750 - 0 Ma ? DO I have missed something or please state the reasons behind this particular choice.

It has been shown that the COPSE / GEOCARB models (of which ours is a simplified version) do a reasonable job of predicting CO₂/temperature for 400-0 Ma, and our degassing rates for this period are similar to previous estimates, so there is little change. However, it does seem sensible to plot the whole Phanerozoic so we have now done this.

Please see attached file for more comments

We have been through the file and made the majority of the suggested changes. We appreciate the efforts of the reviewer here. The changes listed below have either been made subject to some alterations, or have not been made for the following reasons (line numbers here refer to the location of the reviewer’s comment in the original pdf):

Line 40: We think the sentence is useful in a multi-disciplinary journal, so we have kept it in

Line 78: We think it is more reasonable to leave this data in figure 2 where it can be compared to the other reconstructions

Line 81: We have added the solar flux evolution to figure 4 instead of figure 1 as suggested. We feel it is more useful here.

Line 88: we tried switching these paragraphs as suggested but it didn't read as well in our opinion

Line 131: we have now cited Van der Meer et al. (2017) and noted that both our paper and this work support the same general trend in ridge generation rates. We feel this is enough of a comparison for the present work, and that further discussion (which necessitates getting into the strontium system) would detract from the main point of the paper.

Line 161: This version of COPSE does not have the 'NV' parameter (controlling mafic weathering) as it does not distinguish between different silicate lithologies. We have added a note on this to the model section in the SI.

Line 312: The reviewer suggests adding the supercontinent cycle and LIPs to figure 1, but also criticised our comparisons of the supercontinent breakups, which we have subsequently removed from the text. We have therefore shown these in the SI instead.

Line 336: We have tried combining figure 3a and 3b but it does not work nicely, as they measure different things. We have updated our inverted sea level curve following the reviewer's advice.

Reviewer #3 (Remarks to the Author):

The authors aim to explain why there has not been a return to a Cryogenian Snowball Earth climate during the Phanerozoic. This is argued done two-ways, via global length of subduction zones from a plate tectonic reconstruction and via sea-level inversion. As such the application of these methods this far back in time is novel and the paper potentially provides a key insight in paleo-climate of the Neoproterozoic-Phanerozoic boundary which is original and will influence the thinking in the field.

Neither method is perfect and are hindered by a lack of intra-oceanic subduction data and respectively, due the fact that paleo-sea-level is influenced by more than factors just ocean spreading rates. More discussion on these limitations should be included.

In response to this and other comments we have added text on the limitations of the inversion method, and on the uncertainty and limitations of the PALEOMAP method.

[ms line 96, 113, 136, 220]

The paper would also benefit from the incorporation of more recent sea-level curves, and comparisons to plate tectonic activity from other plate tectonic models that also go back to this far back in time.

We have now updated the sea level curves used in the paper, and have included a comparison to the most recent kinematic model of Precambrian plate motions (Meredith et al. 2017. Gondwana Research), which also includes some computations for the Phanerozoic. There is good agreement on the total length of subduction zones between methods.

[ms line 116, 133, 150]

Figure 4 should include a view on the entire Phanerozoic, providing evidence that the methodology works at times where more data is available.

Following this and other comments we have extended the model to the present day and compared to estimated CO₂ thresholds for glaciation.

REVIEWERS' COMMENTS:

Reviewer #1 (Remarks to the Author):

Overall, the authors have revised appropriately and I am generally satisfied.

However, a few points that I did not catch in the first review.

1) Given that there are no page limits in NC, the authors should briefly discuss in the text how they parametrized CO₂ production in through arcs via subduction. Perhaps this is there, but it is not very clear. First, how much carbonate is actually subducted and how does this change through time? Presumably in the Neoproterozoic, much of the carbonate was deposited on continental shelves and did not get subducted. Only later in Earth's history might there be sufficient carbonate subducted, and then, according to Dasgupta's (see RiMG volume) work, it's not obvious how much slab carbonate actually decarbonates. Similar discussions have been had by Kerrick. And even if carbonate subducts and then decarbonates, it's not obvious if this results in net CO₂ production or is a zero sum game, even if you have more subduction zones. The way out of this really is the growing reservoir of continental carbonate and with more subduction zones, particularly continental arcs, you tap more and more of this as you proceed into the Phanerozoic. So good that the authors incorporated this more. Nevertheless, still worth considering some of the points I made above.

2) minor points. It would be nice to plot CO₂ production in addition to pCO₂ with time. It would be nice to also explain briefly in simple terms (I know it comes out of the model, but for most, that will be blackbox) why with higher inputs of CO₂ in neoproterozoic times, pCO₂ is actually higher than today. Is the efficiency of weathering not as strong then?

3) minor points. Fig 4b and elsewhere need better captions. For example, in 4b, the caption doesn't tell us what Wx0.5 and Wx0.2 is.

4) I still have questions about using the Haq and Vail curves to literally. But if the authors are convinced, then go for it.

I don't need to re-review and trust the authors to make the appropriate revisions.

Reviewer #2 (Remarks to the Author):

This is the second times I have this paper. The authors have done a perfect job when modifying to account for reviewers comments. I think the paper is now acceptable as it is.

Reviewer #3 (Remarks to the Author):

The authors have incorporated most of my suggestions or have adequately addressed those in their response. The only remaining point that I would recommend changing is replacing Figure 3C, with Figure S6B (and associated caption and discussion) in order to more clearly show the updated data and references following the sea level inversion approach.

Final comments from reviewers.

Our responses are in blue.

Reviewer #1 (Remarks to the Author):

Overall, the authors have revised appropriately and I am generally satisfied.

However, a few points that I did not catch in the first review.

1) Given that there are no page limits in NC, the authors should briefly discuss in the text how they parametrized CO₂ production in through arcs via subduction. Perhaps this is there, but it is not very clear. First, how much carbonate is actually subducted and how does this change through time? Presumably in the Neoproterozoic, much of the carbonate was deposited on continental shelves and did not get subducted. Only later in Earth's history might there be sufficient carbonate subducted, and then, according to Dasgupta's (see RiMG volume) work, it's not obvious how much slab carbonate actually decarbonates. Similar discussions have been had by Kerrick. And even if carbonate subducts and then decarbonates, it's not obvious if this results in net CO₂ production or is a zero sum game, even if you have more subduction zones. The way out of this really is the growing reservoir of continental carbonate and with more subduction zones, particularly continental arcs, you tap more and more of this as you proceed into the Phanerozoic. So good that the authors incorporated this more. Nevertheless, still worth considering some of the points I made above.

We have added text to this affect and a citation to Dasgupta's article. We explain that our model assumes that CO₂ release is directly proportional to material subduction rate, and explain the uncertainties and caveats that this entails.

2) minor points. It would be nice to plot CO₂ production in addition to pCO₂ with time. It would be nice to also explain briefly in simple terms (I know it comes out of the model, but for most, that will be blackbox) why with higher inputs of CO₂ in neoproterozoic times, pCO₂ is actually higher than today. Is the efficiency of weathering not as strong then?

CO₂ production follows the subduction zone length which is already plotted, so we do not see it essential to add it again, especially as Figure 4 is already complicated. But we have added further explanation on why the pCO₂ is higher than today in the Neoproterozoic even though degassing is similar: a combination of weaker weathering without land plants and a lower solar constant.

3) minor points. Fig 4b and elsewhere need better captions. For example, in 4b, the caption doesn't tell us what Wx0.5 and Wx0.2 is.

This has been amended.

4) I still have questions about using the Haq and Vail curves to literally. But if the authors are convinced, then go for it.

We agree about the uncertainty in interpretation of these curves, but this interpretation is widely used, and it is therefore worthwhile to show it does not falsify our work.

I don't need to re-review and trust the authors to make the appropriate revisions.

Thanks.

Reviewer #2 (Remarks to the Author):

This is the second times I have this paper. The authors have done a perfect job when modifying to account for reviewers comments. I think the paper is now acceptable as it is.

Thanks.

Reviewer #3 (Remarks to the Author):

The authors have incorporated most of my suggestions or have adequately addressed those in their response. The only remaining point that I would recommend changing is replacing Figure 3C, with Figure S6B (and associated caption and discussion) in order to more clearly show the updated data and references following the sea level inversion approach.

We have made these changes.